# Enhancing the Biodiesel Production Potential of *Synechococcus elongatus* and *Anabaena* Cyanobacterial Strain Isolated from Saline Water Using Different Media Composition and Organic Carbon Sources

**Jeevitha Parthiban** [1] and **Ranjitha Jambulingam** [2,*]

[1] School of Advanced Science, Vellore Institute of Technology, Vellore 632014, India
[2] CO$_2$ Research and Green Technologies Center, Vellore Institute of Technology, Vellore 632014, India
* Correspondence: ranjitha.j@vit.ac.in

**Abstract:** In the present study, *Synechococcus elongatus* and *Anabaena*, two cyanobacterial species were cultured using different media conditions such as ASN III, modified ASN III, BG-11, and BBM for the enrichment of biomass and lipid productivity. The experimental result clearly shows that BG 11 was the efficient and cost-effective medium for both the isolated cyanobacterial species such as *Synechococcus elongatus* and *Anabaena*. The influence of organic carbon sources on biomass and lipid productivity of the selected cyanobacterial species were studied when cultivated in a BG-11 medium using different organic carbon sources such as sucrose, glucose, sodium acetate and glycerol under mixotrophic conditions. Based on the experimental results, the isolated cyanobacterial strain *Synechococcus elongatus* and *Anabaena* showed an enriching effect on lipid production under mixotrophic conditions, but whereas *Synechococcus elongatus* showed a significant effect three times greater lipid productivity compared with *Anabaena* cyanobacterial strain, by the addition of glycerol as a supplement to the culture media.

**Keywords:** *Synechococcus elongatus*; *Anabaena*; biomass production; lipid production; biodiesel; carbon sources

## 1. Introduction

Microalgae are microscopic in size that can be found in both freshwater and seawater [1,2]. More than 30,000 different species have already been separated and identified, and they can be divided into prokaryotic cyanobacteria (or) eukaryotic microorganisms [3]. These microorganisms carry out photosynthesis as a natural mechanism for reducing the carbon dioxide concentration in the atmosphere. The short generation time and exponential growth of microalgae in a favorable environment are the main characteristics of microalgae [4]. Generally, three types of cultivation techniques were used for the growth of microalgae species, and they are autotrophy, heterotrophy, and mixotrophy cultivation techniques. These cultivation techniques were used based on environmental factors that are required for their metabolism [5,6]. The majority of autotrophic cultivation is carried out in the open pond and closed photobioreactor (PBR) systems. Generally, organic compounds are used as a carbon source, which is required for energy production in heterotrophic cultivation [7]. Under mixotrophic metabolism, a two-stage growth system, the first stage is a heterotrophic process, when the organic carbon content is high, and the second stage is autotrophic, which starts the initiation of photosynthesis. Microalgae have the tendency to capture CO$_2$ and utilize it as an energy source for biomass and lipid production [7–9]. Most of the microalgae can utilize flue gas emissions from the environment for their nutritional needs [10] and remove pollutants from wastewater [11,12]. Furthermore, microalgal biomass has several applications, such as phytomedicines, bioethanol, biomethane, biohydrogen, biodiesel, etc. In most cases, microalgae containing high lipids content can be used for the production

of biodiesel in the majority of developed nations [13]. During the extraction of the fatty acids, the microalgal residue was used for the production of bio-methane, bio-ethanol and bio-diesel [13–21]. It is a source of amino acids, pigments, and vitamins in the cosmetic and pharmaceutical industries [22,23]. Consequently, microalgae are essential in biofertilization, promoting the growth of various plants. The present research work is mainly focused on the growth optimization study of the isolated cyanobacterial species *Synechococcus elongatus* and *Anabaena* from seawater using different carbon sources in the presence of different media conditions.

## 2. Materials and Methods

### 2.1. Isolation and Screening of Cyanobacterial Strains

The seawater samples were collected from Keelakkarai, Pamban coastal location of the Ramanathapuram district. To keep the cyanobacterial cultures active, the collected seawater sample was put into a sterile bottle and stored at $30 \pm 2$ °C. These samples were cultured in ASN III, BG-11, modified ASN III and BBM medium and kept the cultures for incubation for about seven days. After a week, the culture begins to use the medium for development, and the conical flasks start to show filamentous growth. In modified ASN III, BG-11, ASN III and BBM medium, cyanobacteria were grown in the medium. In direct isolation, the selected colonies were picked from the ASN III, BG-11, modified ASN III and BBM medium and viewed under a compound microscope. The water sample was serially diluted using distilled water under a laminar airflow chamber for serial dilution and streak plate method. The various concentrations ranged between $10^{-9}$, $10^{-8}$, $10^{-7}$, $10^{-6}$, $10^{-5}$, $10^{-4}$, $10^{-3}$, $10^{-2}$, and $10^{-1}$ were diluted and used for further sampling. A total of 1 mL of the stock solution was diluted with 9 mL of distilled water using a micropipette. Each 1 mL of the sample was taken from the previously mixed solution. The selected colonies were inoculated at ASN III, BG-11, modified ASN III and BBM medium and kept at $30 \pm 2$ °C. The pure colonies were cultured in ASN III, BG-11, modified ASN III and BBM medium in a rotary shaker at 250 rpm under photoperiod at RT $30 \pm 2$ °C. Finally, the obtained pure cultures were used for mass culturing of the isolated cyanobacterial strains.

### 2.2. Mass Culturing of Isolated Cyanobacterial Strain

The isolated cyanobacterial strain isolated from saline water was cultivated in different culture mediums to analyze the significant impact on growth metabolism and lipid productivity. The four different media used in the photoautotrophic cultivation of the isolated cyanobacterial strains are ASN III, BG11 [24], BBM, and modified ASN III [25].

### 2.3. Influence of Organic Carbon Sources on Lipid and Biomass Production

The impact of various sources of carbon such as glycerol, sucrose, sodium acetate and glucose was added to the isolated cyanobacterial strains and tested in the ASN III media, modified ASN III BG-11 and BBM. In the experiment, 0.5 g/L quantity of different organic sources such as glycerol, sucrose, sodium acetate and glucose under a sterilized medium. The experiments were performed for approximately 16 days at pH 7.3 (30 °C) under fluorescent lamp conditions.

### 2.4. Media Optimization and Analytical Characterization Studies

The optical density at 680 nm (OD680) was used for the measurement of algal biomass (g/L), using a UV-visible spectrophotometer (Hitachi 5300) to measure the growth of microalgae. For the estimation of biomass, a sample of algae in the amount of 50 mL was filtered through a Whatman filter that had been pre-weighed before being dried in an oven at 60 °C until constant weight. The biomass productivity P (mg/L/day) was determined using empirical formula;

$$P = (W_2 - W_1)/T \tag{1}$$

where T = cultivation time; $W_2$ = biomass concentration of the last day; $W_1$ = initial biomass concentration.

### 2.5. Extraction of Lipid from Microalgal Biomass

The lipid was extracted from the isolated cyanobacterial strain by the Folch extraction method [26]. Initially, 500 mg of biomass was treated with 10 mL of chloroform/methanol (2:1 *v/v*), and the mixture was stirred at room temperature for 30 min. The cell debris and supernatant were separated from the mixture by centrifuging it at 6000 rpm for 15 min. This supernatant was rinsed with 1% NaCl solution and agitated in a vortex mixture for 10 min and centrifuged for 15 min at 3000 rpm and the lower chloroform layer with lipid was removed and collected for further processing. The collected organic layer contains lipids, and excess solvent was removed using a distillation process. Using gravimetric, the lipid content was determined. The following equation was used to compute the lipid productivity (mg/L/d):

$$\text{Lipid productivity} = (\text{Lipid content}_t \times W_1 - \text{Lipid content}_o \times W_2)/t \times 1000 \qquad (2)$$

where Lipid content$_o$ = Algal cells' initial lipid concentration; Lipid content$_t$ = the level of lipids on the final day of cultivation.

### 2.6. Evaluation of the Fatty Acid Profile and Fuel Properties

The lipid extracted from the microalgal strain was converted into fatty acid methyl esters using the Folch method. A molar ratio of (6:1) methanol and lipid was treated with 2.5 mL of 1% potassium hydroxide and heated in a magnetic stirrer for about 3 h at 55 °C. After 3 h, the resulting solution was washed with 15 mL of n-hexane as an organic solvent. The organic layer was separated from the aqueous layer and the organic layer was further distilled using a simple distillation unit to remove the excess of the solvent and finally obtained the product as FAME (Fatty acid methyl esters) [27]. Fatty acids in 200 μL hexane were used for the analysis of a gas chromatograph (Agilent 8890 series) fitted with a Flame Ionization Detector (FID) with a flow rate of 1 mL/min. Fatty acids peaks were identified using a GC analysis of their retention period with an actual standard and were then standardized. As an internal standard, heptadecanoic acid was used. The fuel characteristics of the biodiesel produced from *S. elongatus* species were analyzed using standard protocol [28].

### 3. Results and Discussion

*Synechococcus elongatus* and *Anabaena* are two different cyanobacterial strains isolated from saline water and belong to the families *Synechococcaceae* and *Nostocaceae* and class *Cyanophyceae* respectively. The two cyanobacteria strain morphology was identified using an optical microscope. The individual single colony cells ranged from size 2 to 10 μm. Microalgal cells are spherical, unicellular, and green in color. The two cyanobacterial species, *Synechococcus elongatus* and *Anabaena*, were cultivated in 500 mL Erlenmeyer flasks (conical flasks) with 250 mL of each of four media: ASN III, modified ASN III BG-11, and BBM. The findings showed that, in all media, the isolated microalgal species exhibited a lag time of 2 days and an exponential phase of 6 days and finally reaching the stationary phase as shown in Figure 1a,b. Consequently, the selected microalgal strain biomass dry weight (OD) does not remain constant with optical density when varying the growth media conditions. Figure 2 shows the dry-weight biomass concentration of different media compositions of the isolated microalgal strain at the stationary phase. Among the two microalgal strains, modified ASN III (1.45 ± 0.07 g/L), ASN III (1.39 ± 0.06 g/L), BBM (1.59 ± 0.07 g/L) and BG-11 (1.65 ± 0.08 g/L) and showed significant biomass concentrations were found in *Synechococcus elongatus*, respectively. In BG-11, *Anabaena* (1.55 ± 0.07 g/L) exhibited the highest biomass concentrations. This is due to the increased concentrations of nitrogen which are accelerating biomass growth [29]. Modified ASN III exhibited the highest nitrogen concentration, which resulted in reduced biomass concentration [29].

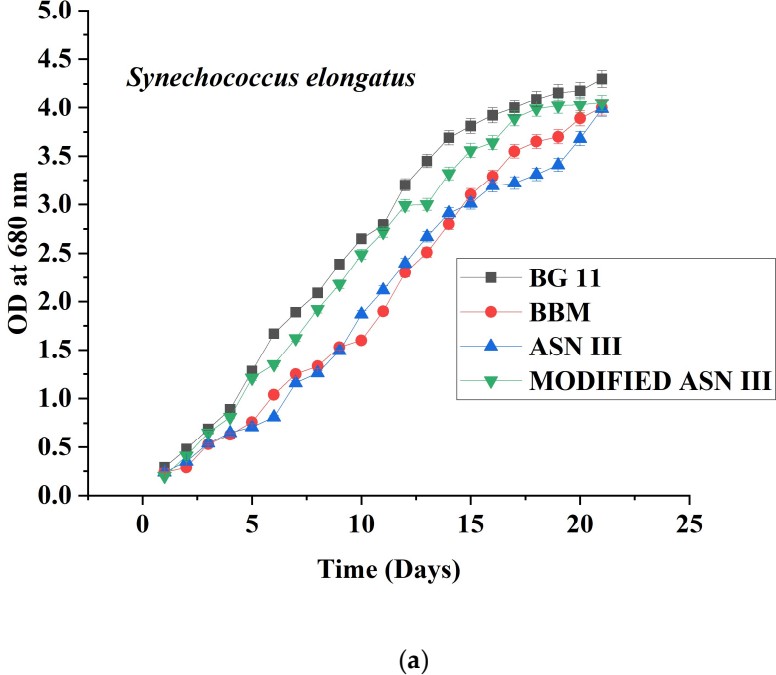

(**a**)

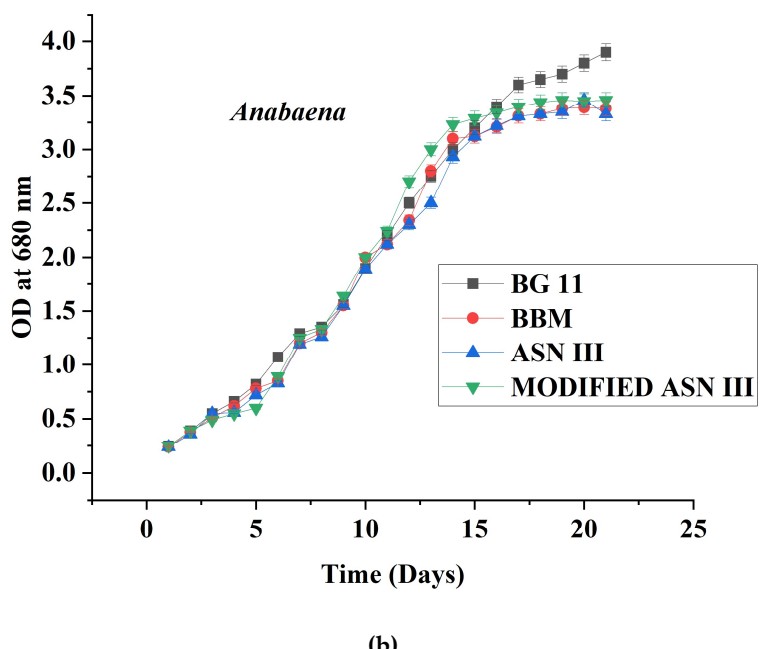

(**b**)

**Figure 1.** Growth curves of (**a**) *Synechococcus elongatus* and (**b**) *Anabaena* under different media conditions.

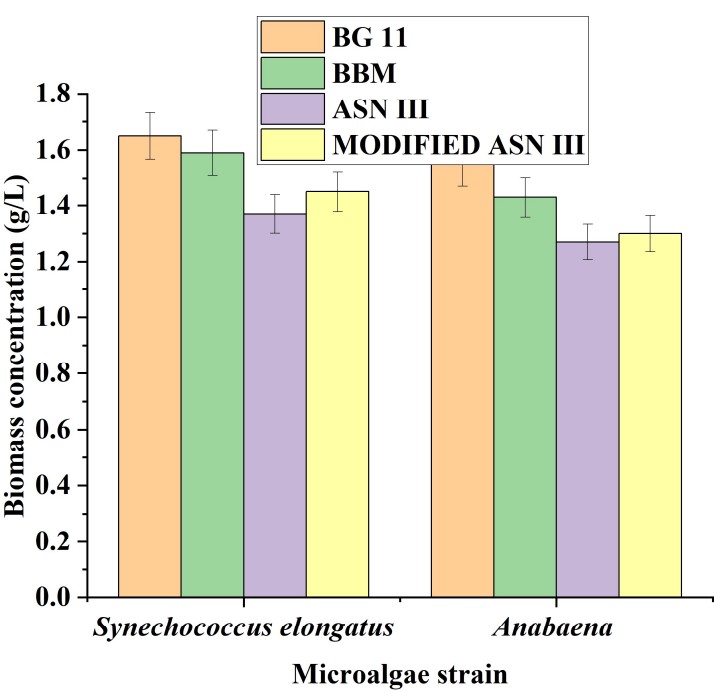

**Figure 2.** Biomass DW concentration comparisons of *Synechococcus elongatus* and *Anabaena* under different media conditions.

The percentage of lipids to biomass dry weight and lipid content were recorded under different medium conditions. A significant difference in lipid content between the isolated strains and culture conditions and biomass concentration was observed. Figures 3 and 4 illustrate the lipid contents and productivity of two isolated microalgal strains under different culture mediums. In *Synechococcus elongatus*, BBM media showed the highest lipid content at $15.90 \pm 0.75\%$, whereas *Anabaena* at $13.03 \pm 0.41\%$ showed comparatively less than *S. elongatus* in BBM. In the ASN III growth medium, the lowest level of lipid content for both the isolated microalgal strains. *Synechococcus elongatus* exhibits the highest lipid productivity at $13.10 \pm 0.6$ mg/L/d, followed by *Anabaena* at $11.28 \pm 0.61$ mg/L/d showed comparatively less than *S. elongatus*. The selection of culture media is influenced by a number of variables, including the desired product, medium cost and growth rate. For the biomass growth and lipid productivity of different microalgae, nutrient plays a vital role in the growth metabolism. Under nitrogen-starvation situation causes higher fat accumulation supports these findings [30–33]. The selected media ASN III and modified ASN III exhibited the lowest lipid content and productivity on average, followed by BBM and BG-11, while ASN III exhibited the highest on average, according to a comparison of culture media. When microalgae are grown under unfavorable conditions, including lacking nutrients, their lipid content often rises [30–32]. Li et al. (2012) reported that nitrogen content can increase the lipid percentage of some microalgae species by up to 30%. In comparison to the BG-11 medium, nitrogen-rich showed enhanced lipid productivity, whereas BBM medium showed significant lipid content and productivity due to its lower phosphate and nitrogen concentrations [29].

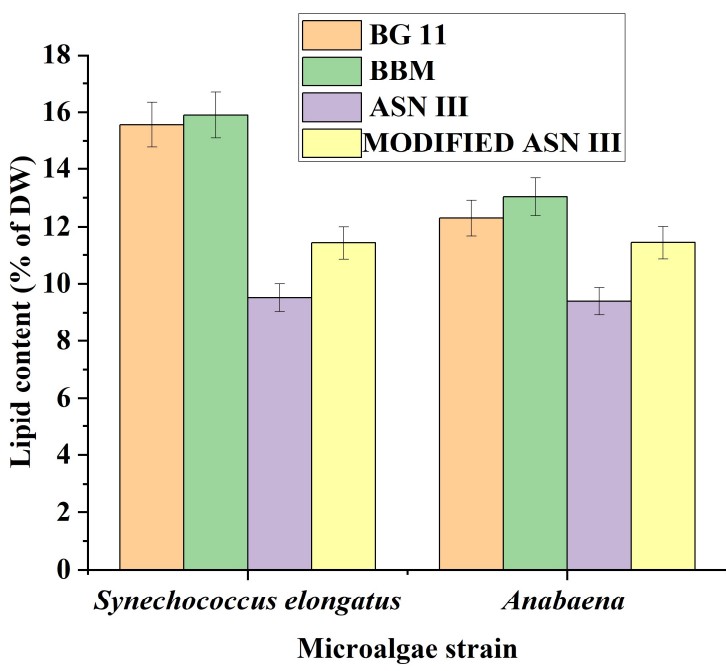

**Figure 3.** Lipid content comparisons of *Synechococcus elongatus* and *Anabaena* under different media conditions.

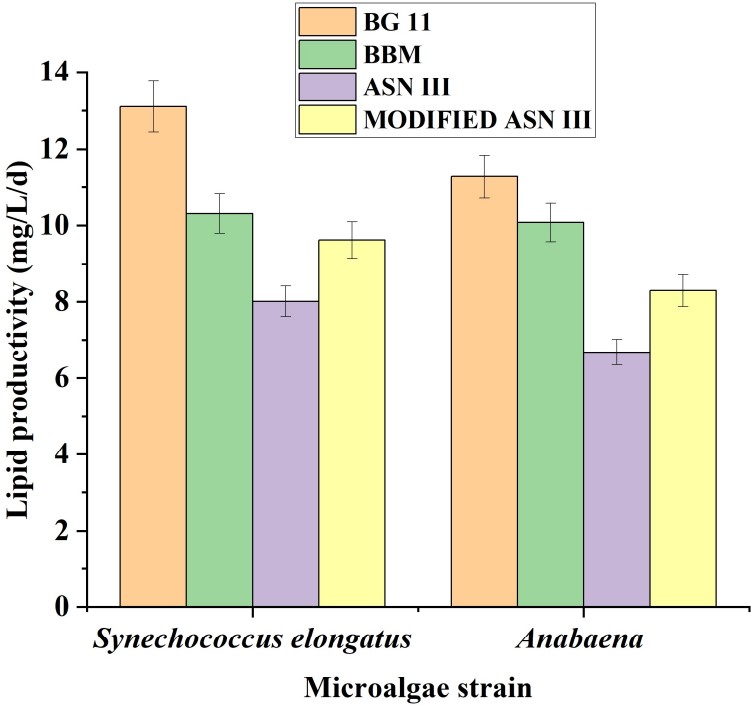

**Figure 4.** Lipid productivity comparisons of *Synechococcus elongatus* and *Anabaena* under different media conditions.

### 3.1. Impact of Organic Carbon Sources on Lipid and Biomass Productivity

Two cyanobacterial strains, *Synechococcus elongatus* and *Anabaena*, were cultivated in an optimized BG-11 medium with various organic carbon sources at room temperature under a 16:8 light: dark cycle [34]. Different organic carbon sources, such as glycerol, sucrose, glucose and sodium acetate, were used with a fixed concentration of (0.5 g/L) to enhance the lipid and biomass production of the selected cyanobacterial strains. Under

the mixotrophic growth of two cyanobacterial strains, glucose was shown to be the best organic carbon source, such as glycerol, sodium acetate, and sucrose. According to the study, when glucose was added, *Synechococcus elongatus* and *Anabaena* produced their highest levels of biomass at 2.15 and 1.85 g/L, respectively, during the stationary phase Figure 5a,b. Similarly, in the medium supplemented with glucose along with the isolated microalgal strains and achieved the highest biomass productivity was 274.46 mg/L/day for *Synechococcus elongatus* and 240 mg/L/day for *Anabaena*, respectively. These results show glucose is an effective carbon source for enhancing microalgal biomass productivity [35–38].

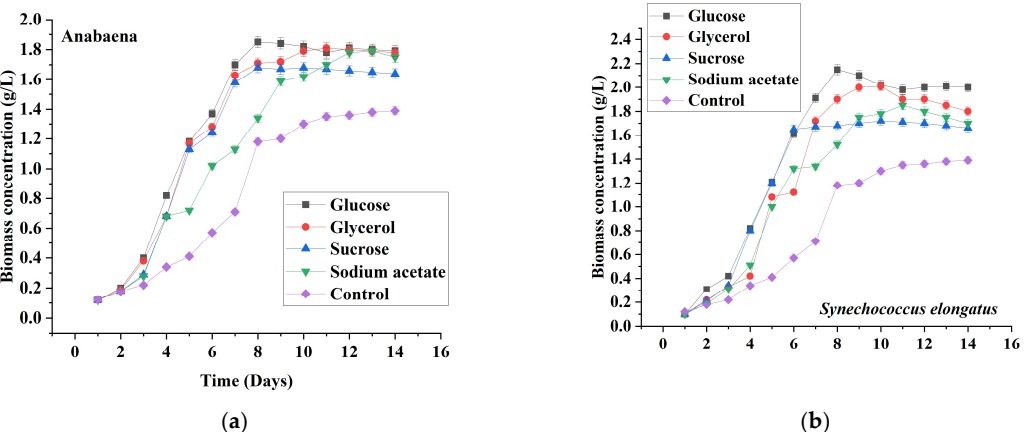

(a)  (b)

**Figure 5.** Biomass growths of (**a**) *Synechococcus elongatus* and (**b**) *Anabaena* under different carbon sources.

Out of the four carbon sources, glycerol showed significant lipid content in *Synechococcus elongatus*, but *Anabaena* showed lesser lipid content compared with *Synechococcus elongatus* (Figure 6). *Synechococcus elongatus* and *Anabaena* both produce 24.12% and 17.19% of the lipid, respectively. In the glycerol medium, the lipid yield was analyzed and found to be 491.04 mg/L for *Synechococcus elongatus* and 370 mg/L for *Anabaena*. This is because glucose is converted into glucose-6-phosphate into pyruvate and enters the TCA cycle, where it is used to produce pyruvate into ATP via the mitochondrial oxidative phosphorylation pathway [39,40]. The maximal lipid productivity for *Synechococcus elongatus* and *Anabaena* was found to be 50.17 mg/L/d and 35.89 mg/L/d, respectively, in the glycerol medium (Figure 7).

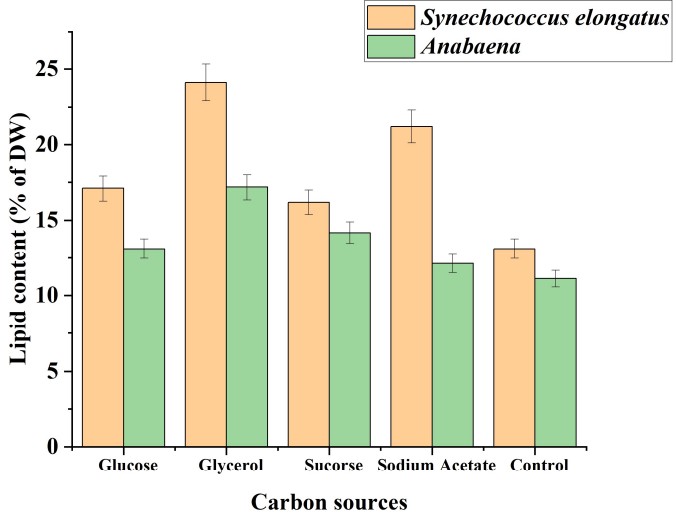

**Figure 6.** Lipid content comparisons of *Synechococcus elongatus* and *Anabaena* in four different carbon sources.

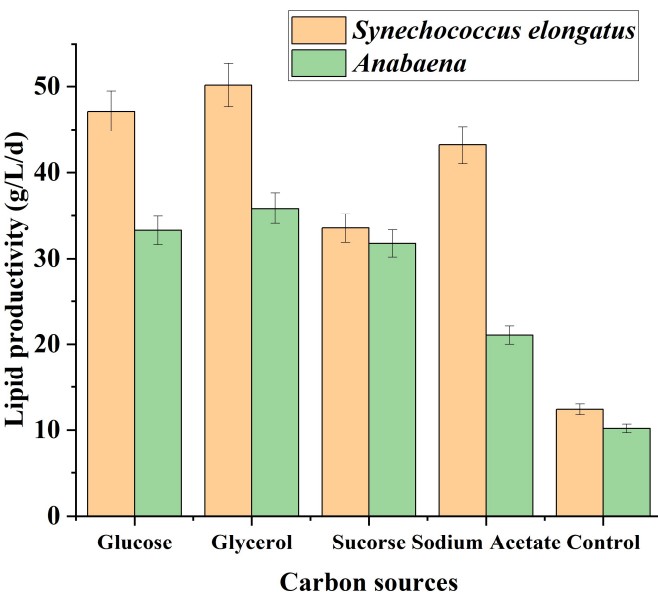

**Figure 7.** Lipid productivity comparisons of *Synechococcus elongatus* and *Anabaena* in four different carbon sources.

*3.2. Analytical Characterization of the Lipids and Biodiesel Obtained from the Isolated Cyanobacterial Strains*

The lipids obtained from the isolated microalgae (*Synechococcus elongatus* and *Anabaena*) were analyzed in a GC instrument to identify the fatty acid content, as shown in Table 1. Based on the GC results, *Synechococcus elongatus* contains higher saturated fatty acid levels (33%) than *Anabaena* (30%). The extensive survey suggests that the perfect biodiesel should contain a certain quantity of saturated and polyunsaturated fatty acids to sustain its strong oxidative stability and overwhelm the cold flow model [41,42]. Monounsaturated fatty acids, such as oleic acid (18:1) and palmitoleic acid (16:1), are essential for achieving the optimal balance between oxidative stability and cold flow [41,42]. Reports indicate that biodiesel blends with a larger proportion of C16–C18 fatty acids have excellent fuel properties [41]. As demonstrated in Table 1, linolenic (18:3), oleic (18:1), palmitic (16:0), linoleic (18:2), palmitoleic (16:1) and stearic (18:0) acids constitute the primary components of the fatty acid compositions of the isolated cyanobacterial strains. Lower biodiesel oxidation stability is caused by a higher polyunsaturated fatty acid, according to European standards EN14214 [41]. The lowest concentration of linolenic acid (C18:3) was found in *Synechococcus elongatus* (6.1%), and the highest concentration was found in *Anabaena* (5.49%).

**Table 1.** Fatty acid profile of *Synechococcus elongatus* and *Anabaena*.

| Fatty Acid Composition (%) | *Synechococcus elongatus* | *Anabaena* |
|---|---|---|
| Polyunsaturated fatty acids | 36.1819 | 33.179 |
| Saturated fatty acids | 33.1539 | 32.0591 |
| Monounsaturated fatty acids | 37.1480 | 29.1870 |
| Octadecanoic acid | 4.1937 | 3.7882 |
| Tetradecanoic acid | 2.8932 | 2.1814 |
| Palmitoleic acid | 2.6175 | 2.3389 |
| Linolenic acid | 6.102 | 5.4918 |
| Lauric acid | 1.3172 | 1.1028 |
| Decanoic acid | 1.5241 | 0.8142 |

The biodiesel produced from *Synechococcus elongatus* and *Anabaena* was tested for their fuel properties, as shown in Table 2. The total quantity of biodiesel unsaturation was

analyzed by iodine value. The polymerization of glycerides and the deposit of lubricant in the engine could happen as a result of the biodiesel's higher iodine value [43]. According to European Standard EN 14214, the iodine value of biodiesel should not be higher than 120 g I2/100 g. According to this study, *Anabaena* has the lowest iodine value (88.17), while Synechococcus elongatus has the highest value (93.04). The biodiesel should contain the least cetane value should be 47, 45 and 51, respectively, according to ASTMD6751, European (EN 14214), Australian, and National Petroleum Agency (ANP255) standards. The presence of saturated fatty acids in lipids has a significant impact on these characteristics. The cetane values of *Synechococcus elongatus* and *Anabaena* in this experiment were found to be 57.42 and 53.51, respectively. The oxidative stability must be greater than 6 h, according to EN 14214. The biodiesel produced by the *Synechococcus elongatus* shows oxidative stability at 6.8 h, whereas *Anabaena* shows oxidative stability at 6.1 h. They could point range of ASTM D6751 is −3 to 12 °C, and the pour point range is −15 to 20 °C. For *Synechococcus elongatus* and *Anabaena*, the cloud point ranges are 10.81 and 5.7 °C, respectively, whereas the pour point ranges are 3.18 and 4.11 °C. Calorific value, a measure of the amount of energy released during fuel combustion, was found to be lowest for *Anabaena* (39.11 MJ/Kg) and highest for *Synechococcus elongatus* (40.18 MJ/Kg) [44]. Based on the above fuel properties, the biodiesel produced from the lipids of *Synechococcus elongatus* and *Anabaena* can be used as biofuel in the diesel engine after analysis of their engine characteristics in the future at the industry level.

**Table 2.** Fuel characteristics of *Synechococcus elongatus* and *Anabaena* biodiesel production.

| Fuel Properties | ASTM | *Synechococcus elongatus* | *Anabaena* |
|---|---|---|---|
| Pour point (°C) | −5 to 10 | 3.18 | 4.11 |
| DU (%) | - | 89.91 | 83.11 |
| Higher heating value (MJ/Kg) | 45.5 | 40.18 | 39.11 |
| Cloud point (°C) | −3 to 15 | 10.81 | 5.7 |
| LCSF (°C) | - | 4.99 | 4.77 |
| CFPP (°C) | - | −1.47 | −1.11 |
| Oxidation stability (h) | Min 8 hrs | 6.8 | 6.1 |
| Cetane number | 47–65 | 57.42 | 53.51 |
| Saponification value (mg KOH $g^{-1}$ oil) | 218 | 182.10 | 179.21 |
| Iodine value (g $I_2$100 $g^{-1}$ oil) | 120 | 93.04 | 88.17 |

## 4. Conclusions

The present study concludes that two cyanobacterial species, *Synechococcus elongatus* and *Anabaena,* were isolated from the saline water. The isolated cyanobacterial strain was cultivated in four different culture mediums to enhance biomass and lipid productivity. Out of all four, ASN III, BG-11 modified ASN III, and BBM medium showed significant lipid production in BG11 when supplemented with glycerol. *Synechococcus elongatus* showed higher lipid content compared with the *Anabaena* species in BG-11, growth media supplemented with glycerol under mixotrophic conditions. *Synechococcus elongatus* and *Anabaena* can be potential candidates for the production of biodiesel production based on their fatty acid composition and fuel properties. The biodiesel produced from *Synechococcus elongatus* and *Anabaena* was further tested for engine characteristics to use in diesel engines as a fuel in the future at the industry level.

**Author Contributions:** Methodology, R.J.; Formal analysis, J.P.; Investigation, J.P.; Writing—original draft, J.P.; Supervision, R.J. All authors have read and agreed to the published version of the manuscript.

**Funding:** This research received no external funding.

**Institutional Review Board Statement:** Not applicable.

**Informed Consent Statement:** Not applicable.

**Data Availability Statement:** All data from this study are available.

**Conflicts of Interest:** The authors declare no conflict of interest.

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
