# Peer review of "Enhancing the Biodiesel Production Potential of Synechococcus elongatus and Anabaena Cyanobacterial Strain Isolated from Saline Water Using Different Media Composition and Organic Carbon Sources"

_sustainability, doi:10.3390/su15010870_

Round 1

Reviewer 1 Report

1.      The strains isolated from wastewater are supposed to be robust. When comparing with numerous existing literature, any new findings about their growth parameters in harsh environments?

2.      When it comes to biodiesel, authors should evaluate or predict the fuel properties derived from these microalgae to see if it meets the requirements of biofuels standards

3.      What is the role of catalysts in biodiesel production from the lipids extracted from microalgal strains?

4.      Why have you used four types of medium in the microalgal growth?

5.      Throughout the manuscript, check the annotations properly.

6.      Is it isolated from high salinity-containing wastewater?

7.      What is the lipid percentage that exists in both isolated microalgal strains?

8.      Which fatty acid is the most dominant fatty acid in isolated microalgal strains?

9.      What is the biomass production rate in both isolated species?

10.  How to enhance the biomass and lipid productivity of isolated microalgal strains?

Author Response

  1. The strains isolated from wastewater are supposed to be robust. When comparing with numerous existing literature, any new findings about their growth parameters in harsh environments?

Answer: Compared to the existing literature report, the isolated microalgal strain showed significant growth parameters under harsh environments.

  1. When it comes to biodiesel, authors should evaluate or predict the fuel properties derived from these microalgae to see if it meets the requirements of biofuels standards

Answer: Biodiesel produced from the isolated microalgal species was analyzed to predict the fuel properties and compared with the ASTM Standards.

  1. What is the role of catalysts in biodiesel production from the lipids extracted from microalgal strains?

Answer: The role of the catalysts enhances the rate of reaction thereby increasing the product yield.  

  1. Why have you used four types of medium in the microalgal growth?

Answer:  To compare the feasibility of the four different medium which is suitable for the growth of microalgae. These are the selective medium for the growth of biomass and lipid productivity. Out of four different medium which BG-11 shows good biomass yields and lipid productivity.

  1. Throughout the manuscript, check the annotations properly.

Answer: Checked throughout the manuscript and changed accordingly.

  1. Is it isolated from high salinity-containing wastewater?

Answer: Yes, it is isolated from the high saline water located in the Keelakkarai, Pamban coastal location of the Ramanathapuram district.

  1. What is the lipid percentage that exists in both isolated microalgal strains?

Answer: out of two isolated microalgal strains Synechococcus elongatus contains 24.12%, whereas Anabaena contains 17.19%

  1. Which fatty acid is the most dominant fatty acid in isolated microalgal strains?

Answer: The major fatty acid in the isolated microalagal strain - Mono unsaturated fatty acids and Polyunsaturated fatty acids

  1. What is the biomass production rate in both isolated species?

Answer: Synechococcus elongatus = BG-11 (1.63 ± 0.07 g/L) and Anabaena = BG-11 (1.52 ± 0.03 g/L)

  1. How to enhance the biomass and lipid productivity of isolated microalgal strains?

Answer: Different carbon sources and nitrogen sources are used to enhance the biomass and lipid productivity of two microalgal strains. Carbon source and nitrogen source are the two parameters for the optimization of the growth of cyanobacteria.

Reviewer 2 Report

This is an interesting study and the authors have collected a unique dataset using the cutting-edge methodology. The paper entitled Influence of different media composition and carbon sources on Biodiesel Production potential of Synechococcus elongatus and Anabaena isolated from saline water is generally well written and structured. The manuscript can be accepted with major revision

1.     How many isolates have been screened and identified from industrial wastewater using molecular characterisation study?

2.     Compared other microalagal strain isolated from the wastewater, how this particular microalagal showed significant biomass and lipid productivity?

3.     Out of two different microalgal strain media, which showed significant efficiency in three microalagal isolated strain from industrial wastewater.

4.     Throughout the manuscript check the notations and symbol as per standard format.

5.     Which fatty acids are predominant in the production of biodiesel production?

6.     Emphasis which microalagal strain showed enhanced lipid productivity using carbon and nitrogen sources?

7.     Elaborate on the isolation of microalgal strain from wastewater?

8.     Which medium is the best and most suitable medium to achieve maximum biomass and lipid productivity?

9.     How the catalyst plays a significant in the conversion of lipids into biodiesel?

10.  What is the pollutant tolerance capacity of the isolated microalgal strains?

11.  I recommend to add the Species name for the cyanobacterium Anabaena in the tittle. Furthermore, pls indicate how the authors were identify these isolates?

Author Response

This is an interesting study and the authors have collected a unique dataset using the cutting-edge methodology. The paper entitled Influence of different media composition and carbon sources on Biodiesel Production potential of Synechococcus elongatus and Anabaena isolated from saline water is generally well written and structured. The manuscript can be accepted with major revision

  1. How many isolates have been screened and identified from industrial wastewater using molecular characterisation study?

Answer: Synechococcus elongatus and Anabaena are two microalgal strains were isolated and screened for the molecular characterisation study

  1. Compared other microalagal strain isolated from the wastewater, how this particular microalagal showed significant biomass and lipid productivity?

Answer: From the mixed culture 5 different species were selected based on the growth of the culture. After that five microalgal strains was staining with nile red staining to check the amount of lipid bodies present in the culture. Out of five species two species shows better lipid content because high content will enhance the biodiesel yield

  1. Out of two different microalgal strain media, which showed significant efficiency in three microalagal isolated strain from industrial wastewater.

Answer: Synechococcus elongatus and Anabaena are the two microalgal species isolated from the saline water. Out of two species Synechococcus elongatus showed better yield when compared to Anabaena

  1. Throughout the manuscript check the notations and symbol as per standard format.

Answer: Checked and corrected

  1. Which fatty acids are predominant in the production of biodiesel production?

Answer: Monounsaturated fatty acids and Polyunsaturated fatty acids

  1. Emphasis which microalagal strain showed enhanced lipid productivity using carbon and nitrogen sources?

     Answer: Synechococcus elongatus gives better result in lipid the productivity when compared to Anabaena

  1. Elaborate on the isolation of microalgal strain from wastewater?

     Answer: From the mixed culture 5 different species were selected based on the growth of the culture. After that five microalgal strains was staining with nile red staining to check the amount of lipid bodies present in the culture. Out of five species two species shows better lipid content because high content will enhance the biodiesel yield

  1. Which medium is the best and most suitable medium to achieve maximum biomass and lipid productivity?

     Answer: Out of four different media BG-11 shows best results in biomass and lipid productivity

  1. How the catalyst plays a significant in the conversion of lipids into biodiesel?

Answer: Catalyst will increase the reaction rate thereby increasing the product yield. In my research work I used heterogenous catalysts which are used in the production of biodiesel.

  1. What is the pollutant tolerance capacity of the isolated microalgal strains?

     Answer: The isolated microalgal strain has a significant tolerant capacity .

  1. I recommend to add the Species name for the cyanobacterium Anabaena in the tittle. Furthermore, pls indicate how the authors were identify these isolates?

     Answer: From the saline water, isolated the microalagal strains and analysed.

Reviewer 3 Report

In this m/s, authors observed the effects of different media conditions like ASN III, modified ASN III, BG-11, and BBM on two cyanobacterial species', Synechococcus elongatus and Anabaena,  biomass and lipid productivity. The influence of organic carbon sources on biomass and lipid productivity was discussed in detail. Overall the manuscript presents results that are potentially interesting for lipid production. However, there are several aspects that have not been discussed in detail or missing. 

1. Authors discussed effect of different organic carbon sources on biomass and lipid productivity; why they didnot observed effect of initial concentration of given organic carbon sources?

2.  When authors distinguished the effect on  biomass and lipid productivity from different media conditions like ASN III, modified ASN III, BG-11, and BBM, they just attributed the difference to the nitrogen content. If this difference of different media conditions just exists in  the nitrogen content, are there others?  Please list the component in detail of the mediium used in this study.

3. Some more recent references should be listed and compared.

Author Response

In this m/s, authors observed the effects of different media conditions like ASN III, modified ASN III, BG-11, and BBM on two cyanobacterial species', Synechococcus elongatus and Anabaena,  biomass and lipid productivity. The influence of organic carbon sources on biomass and lipid productivity was discussed in detail. Overall the manuscript presents results that are potentially interesting for lipid production. However, there are several aspects that have not been discussed in detail or missing. 

  1. Authors discussed the effect of different organic carbon sources on biomass and lipid productivity; why they did not observe the effect of the initial concentration of given organic carbon sources?

 Answer: effect of initial concentration is mentioned in the biomass and lipid productivity.

  1. When authors distinguished the effect on biomass and lipid productivity from different media conditions like ASN III, modified ASN III, BG-11, and BBM, they just attributed the difference to the nitrogen content. If this difference of different media conditions just exists in the nitrogen content, are there others?  Please list the component in detail of the medium used in this study.

Answer:  Carbon sources: Glucose, glycerol, sucrose, sodium acetate

  1. Some more recent references should be listed and compared.

Answer: Cited the important references in the manuscript and compared.

Round 2

Reviewer 2 Report

The article is better than a previous version and its may be consider for publication in the current status

Reviewer 3 Report

The authors have addressed most of the points raised by the reviewers. The revised version is acceptable for publication.